



**A measurement and model study on ozone characteristics in marine air at a remote**
**island station and its interaction with urban ozone air quality in Shanghai, China**
Yixuan Gu[a,b], Fengxia Yan[c], Jianming Xu[a,b,*], Yuanhao Qu[a,b], Wei Gao[a,b]
[a]Shanghai Typhoon Institute, Shanghai Meteorological Service, Shanghai 200030, China
[b]Shanghai Key Laboratory of Meteorology and Health, Shanghai Meteorological Service,
Shanghai 200030, China
[c]Meteorological Center of Traffic Management of East China, Shanghai 2000135, China
Corresponding to: Dr. Jianming Xu (metxujm@163.com)
Keywords: Ozone in oceanic air, Urban Plume, Coastal city air pollution, Shanghai





**Abstract**

To understand the characteristics and changes of baseline ozone ($O_3$) in oceanic air in
East China, a six-year measurement of $O_3$ concentration was conducted from January 1
2012 to September 15 2017 at a remote offshore station located on the Sheshan Island
(SSI) near the megacity of Shanghai. The observed monthly mean $O_3$ concentrations at
SSI ranged from 33.4 to 61.4 ppbv during the study period, which were about 80% and 12%
higher, respectively than those measured at downtown and rural sites in Shanghai.
Compared to the remarkable $O_3$ increases observed at urban and rural sites in Shanghai,
observed $O_3$ concentrations at SSI exhibited statistically insignificant increasing changes
(1.12 ppbv yr$^{-1}$, α>0.10) during the observation period, suggesting less impacts of
anthropogenic emissions on $O_3$ levels in oceanic air. In addition, an insignificant
decreasing change (-0.72 ppbv yr$^{-1}$, α>0.10) was detected in $O_3$ concentrations at SSI in
September and October when the influence of regional transport was minimum
throughout the year, providing a good proxy to study the baseline oxidation capacity of the
oceanic atmosphere. City plumes from Shanghai usually carried higher levels of $NO_x$,
resulting in decreased $O_3$ concentrations at SSI during southwesterly and westerly winds.
However, In MAM (March–May) and JJA (June–August), due to the enhanced oxidation of
oxygenated volatile organic compounds, $O_3$ could be continuously produced during
daytime in aged city plumes, resulting in elevated $O_3$ concentrations transported to SSI.
The impacts of the offshore $O_3$ on $O_3$ levels in Shanghai are quantified during an easterly
wind dominant episode (September 1–30, 2014) using the WRF-Chem model. Sensitivity
results suggest that $O_3$ in the oceanic air inflows can lead to 20–30% increases in urban



$O_3$ concentrations, which should be crucially considered in dealing with urban $O_3$ pollution
in large coastal cities like Shanghai.



## 1 Introduction

Ground-level ozone ($O_3$) is a harmful photochemical oxidant detrimental to air quality, human health and land ecosystems (Yue and Unger 2014; Monks et al., 2015; Li et al., 2019a). High ambient $O_3$ has been proved to increase the risks of respiratory and cardiovascular mortality (Goodman et al., 2015) and enhance the greenhouse effect (IPCC, 2013). In recent years, $O_3$ pollution has drawn increasing attention in China, since $O_3$ pollution is getting worse in spite of the implementation of Chinese Clean Air Action Plan. In contrast to the 28-40% decreases in $PM_{2.5}$ (fine particulate matter; diameter ≤ 2.5 μm) levels, the observed daily maximum 8-h average (MDA8) $O_3$ concentrations show increasing rates of 1–3 ppb $yr^{-1}$ in summer in megacities over eastern China during 2013–2017 (Li et al., 2019b). To address the underlying causes of the increasing $O_3$ pollution has become an urgent issue that triggers lots of discussions based on observational and model studies worldwide.

Observational and model studies indicated that the elevated $O_3$ levels in urban and rural areas in eastern China were strongly related to the changes in anthropogenic emissions of $O_3$ precursors (Ma et al., 2016; Lu et al., 2018; Li et al., 2019b; Gu et al., 2020). Since the $O_3$ formation was reported to be under volatile organic compound (VOC) limited regime in most Chinese megacities (e.g. Beijing, Shanghai, and Guangzhou), the sharp decreases in nitrogen oxides ($NO_x=NO+NO_2$) emissions combined with slight increases in VOC levels were suggested to be main causes of the observed enhancement of $O_3$ concentrations in East China (Gao et al., 2017; Xu et al., 2019). In remote areas, changes of baseline $O_3$ also exhibit sensitive responses to human activities (Vingarzan,





2004; Meng et al., 2009; Wang et al., 2009; Lin et al., 2015). Based on 14-year
observations, Wang et al. (2009) pointed out that enhanced pollution flow from the upwind
coastal regions contributed to most of the observed increases in $O_3$ concentrations during
1994–2007. And the increase in background $O_3$ likely made a strong contribution (81%) to
the increasing rate of $O_3$ in urban Hong Kong. To understand the background $O_3$ changes
and its response to human activities are thus necessary in developing long-term
strategies to mitigate local $O_3$ pollution. However, compared to the intensive field studies
in polluted cities and surrounding rural regions, continuous observations of $O_3$ at
representative background sites are relatively limited in China (Wang et al., 2017).

To better understand the characteristics of the background $O_3$ changes in mainland

China, the China Meteorological Administration (CMA) started to conduct continuous
measurements of surface $O_3$ at several regional background stations (e.g. Shangdianzi,
Linan, and Longfengshan) since 2005. Over 10-year records from the three sites and
Waliguan, a baseline Global Atmospheric Watch (GAW) station in Tibetan Plateau region,
exhibited different increases in background continental $O_3$ concentrations especially
during daytime in China (Lin et al., 2008; Xu et al., 2008; Meng et al., 2009; Ma et al.,
2016; Xu et al., 2016). The detected positive trends of surface $O_3$ were in a range of 0.24–
1.13 ppbv $yr^{-1}$, suggesting enhanced atmospheric oxidation capacity response to the rapid
development of urbanization and industrialization in the past decades. In addition to the
changes in background $O_3$ in terrestrial stations mentioned above, the characteristics of
baseline $O_3$ at remote marine sites are also important. It is because that large amounts of
$O_3$ pollution events occurred in coastal urban agglomerations in East China (Lu et al.,





2018; Li et al., 2019a, b), affected by both city plumes and oceanic air inflows (Tie et al.,
2009; Shan et al., 2016). For example, model work of Tie et al. (2009) suggested that sea
air masses carried by oceanic inshore air flows aggravated urban $O_3$ pollution in Shanghai
under convergence conditions. Understanding the $O_3$ characteristics in offshore oceanic
regions is therefore an important prerequisite for understanding the land-sea $O_3$
interactions and its impacts on $O_3$ pollution in coastal cities. However, to our knowledge,
studies on the characteristics and changes of $O_3$ in marine air are quite limited in mainland
China since it is very difficult to conduct systematic and continuous observations under
remote oceanic air conditions.

In this report, we present the first relatively long and continuous measurements of $O_3$

conducted on a remote offshore island (Sheshan Island, SSI) from Jan 2012 to Sep 2017
in eastern China. The SSI is located at the confluence of the Yellow Sea and the East
China Sea, covering about 0.4 square kilometers area. Since there are no inhabitants in
the island, the observed $O_3$ is seldom affected by local anthropogenic emissions. The
collected $O_3$ data are used to understand the magnitudes and variabilities of $O_3$ in the
offshore regions and their impacts on the $O_3$ concentrations in coastal city areas. First
shown are the general impacts of regional transport on the remote atmosphere over the
SSI region. Then the diurnal patterns of $O_3$ at SSI are investigated by comparing them
with those observed at a downtown site (XJH) in Shanghai. Multi-year changes of $O_3$
concentrations at SSI are analyzed to examine the overall changes of baseline $O_3$ in
marine air and possible causes. Also analyzed are the impacts of urban plumes on $O_3$
levels in oceanic air in offshore regions. At last, the influence of $O_3$ carried by oceanic air





inflows on urban $O_3$ air quality in Shanghai is assessed using the Weather Research and
Forecasting model coupled with Chemistry (WRF-Chem).
2 Material and methods
2.1 The SSI site and ozone observations
To investigate the characteristics and variabilities of $O_3$ in marine air and their interactions
with urban air quality in coastal areas, ground $O_3$ concentrations were continuously
measured at SSI site (31.4˚N, 122.3˚E, 73.5 m a.s.l.), which is approximately 75
kilometers away from the east edge of Shanghai city. Figure 1 shows the location of SSI
and the surrounding environment. As mentioned in Sect. 1, there is no resident and tourist
on the island. The observed $O_3$ at SSI site can represent the background $O_3$ conditions in
oceanic air which are seldom contaminated by anthropogenic emissions. Hourly $O_3$ data
was collected during January 1 2012 to September 15 2017, with a capture rate of 89.7%.
$O_3$ was measured using an analyzer from Ecotech, Australia (Model EC9810), which
combined microprocessor control with ultraviolet photometry. The instrument met the
technical specifications for United States Environmental Protection Agency, with a quality
control check every 3 days, filer replaced every 2 weeks and calibration every month.
2.2 Observational data at urban and rural sites in Shanghai
To better understand the characteristics of the offshore $O_3$ in oceanic air at SSI, $O_3$
observations obtained from a downtown site, Xujiahui (XJH) are used for comparisons.
The XJH site is located at downtown Shanghai, approximately 80 km west from the SSI. In
addition, since measurements of $NO_x$, carbon monoxide (CO) and meteorological
parameters (e.g. wind direction and wind speed) were unavailable at SSI, observations



obtained at an adjacent site, Dongtan (DT), are substituted for the investigation. The DT
site was set up in a national nature reserve near the coast of Shanghai, where the
observed pollutant levels have been reported to well reflect the impacts of pollution
transport during the MIRAGE-Shanghai (Megacities Impact on Regional and Global
Environment at Shanghai) field campaign (Tie et al., 2013). Similar to SSI, the DT site is
also little affected by human activities. The obtained observations of meteorology and
pollutants are therefore applied for analyzing the impacts of regional transport on
observed $O_3$ concentrations at SSI. $NO_x$ concentrations were measured with a
chemiluminescent trace level analyzer (TEI; Model 42iTL), with detection limit of 0.025
ppb. CO concentrations were measured by the Model 48iTL Enhanced CO analyzer,
based on gas filter correlation technology. The wind speed and wind direction were
measured by using a DZZ4 Automatic Weather Station certificated by the China
Meteorological Administration. The geographical locations and surrounding environment
of XJH, DT, and SSI are displayed in Fig. 1.
2.3 The WRF-Chem model
We simulate $O_3$ using the regional chemical transport model WRF-Chem (version 3.8,
https://www2.acom.ucar.edu/wrf-chem), collaboratively developed through efforts of
several institutes, such as the National Center for Atmospheric Research (NCAR) and the
National Oceanic and Atmospheric Administration (the National Centers for Environmental
Prediction (NCEP). The model includes on-line calculation of meteorological parameters,
transport, mixing, emission, and chemical transformation of trace gases and aerosols
(Grell et al., 2005). The Regional Acid Deposition Model version 2 (RADM2, Stockwell et





al., 1990) gas-phase chemical mechanism is used for the $O_3$ formation chemistry.
Photolysis rates are calculated by using the fast radiation transfer module (FTUV)
followed those in Madronich and Flocke (1999) and Tie et al. (2003). ISORROPIA II
secondary inorganic (Fountoukis and Nenes, 2007) and the Secondary ORGanic Aerosol
Model (SORGAM) (Schell et al., 2001) schemes are used for aerosol chemistry. Dry
deposition follows the standard resistance-in-series model of Wesely (1989). The major
physical processes employed in the model follow the Lin microphysics scheme (Lin et al.,
1983), the Yonsei University (YSU) planetary boundary layer (PBL) scheme (Hong and
Lim, 2006), the Noah Land surface model (Chen and Dudhia, 2001), and the long-wave
radiation parameterization (Dudhia, 1989).
The model used in this study has a horizontal resolution of 6km×6km, including 150
un-staggered grids in west-east, 150 un-staggered grids in south-north, and 35 vertical
layers extending from the surface to 50 hPa. The domain encompasses Shanghai and
surrounding region, centered at 31.3°N, 121.4°E. The NCEP FNL (Final) Operational
Global Analysis data are used for meteorological initial and boundary conditions, with
lateral meteorological boundary updated every 6 h. Basic chemical lateral boundary
conditions are constrained by a global chemical transport model (MOZART-4, Model for
OZone And Related chemical Tracers, version 4) (Tie et al., 2001; Emmons et al., 2010).
Anthropogenic emissions are derived from the Multi-resolution Emission Inventory for
China (MEIC inventory, http://www.meicmodel.org/; Li et al., 2014) for year 2010. Biogenic
emissions are calculated online using model of emissions of gases and aerosols from
nature (MEGAN2, Guenther et al., 2006).





2.4 Methods for assessing the trend of ozone
The daily mean $O_3$ concentrations are used to examine the overall changes in $O_3$
concentrations during the period 2012–2017, including all time of day with qualified
measurements. The trends are assessed using two nonparametric methods, which are
commonly used to detect trends of non-normally distributed data with seasonality (Xu et
al., 2016).The Mann-Kendall (MK) trend test (Mann, 1945; Kendall, 1975; Gilbert, 1987) is
used to examine the trend significance, and the Theil-Sen trend estimate method (Sen,
1968) is used to estimate the slope of trend, which could also be considered as the rate of
change, during the six-year period. Compared to the linear fitting analysis which require
data to be independent and follow a Gaussian distribution, the non-parametric trend test
methods only need the data be independent (Gocic and Trajkovic, 2013). To determine if
the calculated rate of change is statistically significant, the confidence level of at least 95%
is adopted in the MK trend test, with α value less than 0.05 being considered a statistically
significant trend. The trend significance is examined by comparing the value of a
standardized test statistic Z to that of a standard normal variate at a given significance
level ($Z_\alpha$, α=0.05). If $|Z| > Z_{1-\alpha/2}$, then the dataset is non-stationary, exhibiting either an
increasing or a declining trend; If $|Z| \leq Z_{1-\alpha/2}$, then the dataset is stationary with no
significant trend. Detailed calculation of Z can be referred to Xu et al. (2016).
3 Results and discussion
3.1 Regional transport characteristics at SSI
The observed $O_3$ concentrations at SSI were inevitably influenced by regional transport
depending on the prevailing winds in various seasons. Figure 2 displays the monthly wind





rose diagrams averaged over the period of 2012 to 2017 at DT. As mentioned in Sect. 2.2,
the DT site is a rural site located quite close to SSI. The observed wind speeds and wind
directions could then be applied to deduce the origins of the air mass arriving at SSI in
adjacent region. Generally, observed prevailing winds exhibited distinct seasonal
variabilities which were greatly affected by the East Asian monsoon. In warm seasons
(May-August), the site was predominately influenced by easterly and southeasterly winds,
accounting for 40–50% of the total winds. While in cold seasons (November-February),
the northwesterly and northerly winds became the predominant flows that affected SSI,
accounting for about 45% of total winds. During transitional months (e.g. March, April,
September and October), the dominant winds presented more diversities, with wind
directions dispersedly distributed in all the directions. The observed seasonal variations of
prevailing winds are typical at coastal cities at mid-latitude region (Shan et al., 2016; Xu et
al., 2019), suggesting that air masses arriving at SSI originated from various regions and
could result in different impacts on the offshore atmospheric composition in different
months.

Since CO has a relative long chemical lifetime of a few months, the observed CO

concentrations at DT could be regarded as a consequence of regional transport from
polluted regions (Tie et al., 2009). Figure 3 displays the observed monthly mean CO
mixing ratios under wind directions of north (N), northeast (NE), east (E), southeast (SE),
south (S), southwest (SW), west (W), and northwest (NW) at DT during the 2012–2017
period. Observed CO exhibited relative higher concentrations under SW and W winds in
all months, with mean mixing ratios of 0.44 and 0.56 ppmv, respectively during 2012–2017



(Table 1). The observed high CO mixing ratios suggested that the atmosphere
constituents at SSI could be more affected by regional transport of air pollutants under
SW and W wind conditions. As SSI is located to the northeast of the Shanghai city (Fig. 1),
air masses carried by the SW and W flows usually contain more urban pollutants from
upwind city areas, and those carried by E, SE, and NE flows mostly come from the ocean.
The oceanic air masses are cleaner compared to those from the cities, leading to lower
CO mixing ratio at SSI. For example, observed CO exhibited a mean concentration of
0.23 ppmv under SE wind conditions, which was about 50% lower than that influenced by
W winds. To further examine the impacts of the SW and W winds on the atmosphere
constituents at SSI, Table 2 lists the calculated monthly mean occurrence frequency of the
SW and W winds in separate months during the studied period. The SW and S winds
were most infrequent in September (6.1 %) and October (5.2 %), suggesting that the
atmosphere at SSI during the two months could be less contaminated by pollutants
transported from the city and might be more close to the baseline oceanic air conditions.
3.2 The diurnal pattern of ozone at SSI
Figure 4 displays the monthly mean diurnal variations of $O_3$ at SSI and XJH in different
months during 2012–2017. The observed $O_3$ concentrations at the two sites exhibited
similar seasonal distinctions, with monthly mean values highest (61.4 ppbv for SSI and
35.9 ppbv for XJH) in May and lowest (33.4 ppbv for SSI and 12.5 ppbv for XJH) in
December. Since the $O_3$ formation in urban Shanghai is VOC-limited, observed $O_3$ could
be significantly depressed by large $NO_x$ emissions at downtown site (XJH) (Gu et al.,
2020). In Fig. 4, observed $O_3$ levels at XJH were quite lower than those at SSI in all



months, with mean concentrations of 27.8 and 50.1 ppbv, respectively at XJH and SSI
during the observation period. The observed mean $O_3$ concentration at the rural site of DT
(44.7 ppbv) was also lower than that at SSI, which suggested that the $O_3$ levels in marine
air were higher than the continental $O_3$ at both urban and rural sites. The observed diurnal
patterns of $O_3$ at SSI and XJH were similar to those reported for other sites in eastern
China (Xu et al., 2008; Geng et al., 2015; Gao et al., 2017), exhibiting minimums in early
morning (06:00–08:00 LST) and maximums in the afternoon (13:00–15:00 LST). However,
compared to those at the urban site (XJH), observed amplitudes of $O_3$ diurnal variations
were much smaller at SSI. The diurnal variations of surface $O_3$ can be mainly attributed to
the $O_3$ production through photochemical reactions in the daytime and $O_3$ depression via
NO titration at nighttime (Sillman, 2003). Due to few emissions of $O_3$ precursors ($NO_x$ and
VOCs), the $O_3$ production and depression could be weaker at remote site, resulting in
flatter diurnal cycle of $O_3$ compared to that at polluted urban site.

Since the amplitudes of $O_3$ diurnal variations usually exhibited much smaller values in

background areas compared to those in polluted urban regions, the ratio of daily
maximum $O_3$ concentration ($O_{3\text{-max}}$) to minimum $O_3$ concentration ($O_{3\text{-min}}$) was regarded as
an indicator to identify if the local $O_3$ pollution was significantly influenced by
anthropogenic emissions (Cvitas and Klasinc 1993; Vingarzan, 2004). The $O_{3\text{-max}}/O_{3\text{-min}}$
ratio displayed larger values in polluted regions (Cvitas et al., 1995) and lower values in
less contaminated rural regions. A ratio of about 1.4 suggested that the site could be
regarded as a typical background site (Scheel et al., 1997). In Fig. 4, observed $O_3$
displayed different diurnal variabilities in different months, indicating different influence of





regional transport on $O_3$ levels at SSI. Figure 5 displays the calculated monthly mean
$O_{3\text{-max}}/O_{3\text{-min}}$ ratios at SSI and XJH, respectively during 2012–2017. Generally, the
observed ratios of $O_{3\text{-max}}/O_{3\text{-min}}$ at SSI were much lower than those at XJH in all the
months, suggesting less impact of anthropogenic emissions on $O_3$ levels. The calculated
mean ratios were 3.03 and 5.20, respectively at SSI and XJH, and most of the calculated
values were larger than 4.50 at the urban site. Besides, the ratios presented distinct
seasonal differences at XJH and SSI sites. Higher values were observed in summer,
indicating stronger photochemical production of daytime $O_3$ during June to August. At SSI,
the $O_{3\text{-max}}/O_{3\text{-min}}$ ratio exhibited relatively low values in September and October, ranging
from 1.61–2.35 during the studied period. Since the observed temperature and solar
radiation still exhibited higher values during the two months in Shanghai (Gao et al., 2017),
the observed low $O_3$ diurnal amplitudes should not be attributed to the weakened
photochemical formation of $O_3$ as those in winter. Due to the persistent control of
anticyclone, Shanghai and its neighboring areas are usually dominated by stable weather
conditions in September and October, resulting in more gentle and diversified wind
conditions. During the two months, the occurrences of more polluted SW and W winds
were lowest (6.1% and 5.2%) throughout the year. The corresponding wind speed (2.49
and 2.50 m s$^{-1}$) also exhibited values 20% lower those in other months (Table 2). The
transport conditions led to fewer pollutants transported to the SSI region, which could
explain the observed weak diurnal variabilities of $O_3$ in September and October. The
transport conditions together with $O_3$ response further confirmed that the transport of city
pollutants had minimum impacts on the offshore $O_3$ levels in oceanic air at SSI in





September and October.
3.3 Overall changes of ozone in oceanic air at SSI
Several studies have observed increasing trends of ground-level $O_3$ in metropolitan areas
over eastern China since 2013, suggesting that the $O_3$ increases were mostly attributed to
the $NO_x$ emission reductions (Ma et al., 2016; Gao et al., 2017; Lu et al., 2018; Li et al.,
2019b). However, the $O_3$ changes at remote sites were relatively not well elucidated
during past years. Figure 6a presents the monthly variations of $O_3$ concentrations at SSI
and XJH during the 2012–2017 period. The statistical results of the MK test and Theil–
Sen trend estimate method indicated that observed monthly mean $O_3$ mixing ratios ($O_{3\text{-ave}}$)
exhibited increasing changes at both urban (XJH) and remote sites (SSI) in Shanghai,
with calculated increasing rate of 1.97 and 1.12 ppbv $yr^{-1}$, respectively in XJH and SSI.
Though an overall upward trend of $O_3$ was detected at SSI, the changes were not as
remarkable as those observed at XJH, which could not even pass the MK trend test at the
90% confidence level. The detected increasing trend of $O_3$ in oceanic air at SSI were also
not as remarkable as those observed at remote continental sites (e.g. DT and Lin'an) over
the Yangtze River Delta (YRD) region (Xu et al., 2008; Gao et al., 2017; Gu et al., 2020),
suggesting few influence of anthropogenic emissions on the observed $O_3$ levels in
oceanic air.

As discussed in Sect. 3.1, different prevailing winds led to different transport

conditions at SSI in various months. Comparatively, the observed $O_3$ concentrations at
SSI were least contaminated by the regional transport of air pollutants in September and
October. To further examine the changes of nearly uncontaminated $O_3$, we present the



variations of daily mean surface $O_3$ mixing ratios in September and October at SSI and
XJH, respectively during the six-year period in Fig. 6b. The corresponding mean $O_3$
mixing ratios were 60.9 and 31.3 ppbv, respectively at SSI and XJH. Compared to the
significant increasing changes of $O_3$ (0.59 ppbv yr$^{-1}$, α<0.05) at urban site (XJH), observed
$O_3$ at SSI in September and October exhibited insignificant decreasing changes during
the studied period, with an average rate of -0.72 ppbv yr$^{-1}$. The changes were smaller and
quite different from the overall changes at SSI as well as those detected at XJH, which
further indicated that the observed $O_3$ levels at SSI in September and October should be
seldom influenced by urban plumes, providing a good proxy to study the baseline $O_3$ and
oxidation capacity of background atmosphere in eastern China.
To investigate possible drivers of the observed changes in the least contaminated $O_3$
concentrations in September and October at SSI, Table 3 displays the statistical results of
the MK test and Theil-Sen trend estimate for $NO_x$, CO mixing ratios, temperature, and
wind speed during the 2012–2017 period. Statistically significant upward trends were
detected in $NO_x$ concentrations and wind speed with estimated increasing rates of 0.48
ppbv yr$^{-1}$ and 0.21 m s$^{-1}$ yr$^{-1}$, respectively during the observation period (α<0.05). The
results suggested that increases in surface wind speed might be an important
meteorological driver of the observed decreasing changes in $O_3$ levels at SSI from 2012 to
2017. Since both $NO_x$ and CO levels exhibited different increases, it means more
pollutants could be transported to the island and might resulted in elevated $O_3$
depressions during the period. Figure 6c presents corresponding variations of daytime
and nighttime mean $O_3$ concentrations at SSI. Similarly, insignificant downward changes



were detected in both daytime and nighttime $O_3$ levels, indicating that the diffusion and
depression of $O_3$ might be enhanced at SSI due to elevated wind speeds and $NO_x$
concentrations. Since observations of solar radiation were not available during the study
period, the influence of radiation cannot be analyzed which might also play important role
in determining the overall changes of the observed $O_3$ at SSI.
3.4 Impacts of urban plumes on ozone in oceanic air at SSI
Due to the relatively long residence lifetime (about one month), $O_3$ produced at urban
regions could be transported several hundred kilometers away to downwind areas.
Meanwhile, the urban plumes become more aged with continuous production/depletion of
$O_3$ and its precursors, resulting in non-linear variations of $O_3$ in downwind areas (Geng et
al., 2011; Tie et al., 2009, 2013). Several studies suggested that there tended to be
considerable $O_3$ formations in aged urban plumes in the downwind region of Shanghai
(Geng et al., 2011; Tie et al., 2013). To investigate the impacts of urban plumes on the $O_3$
levels in oceanic air at SSI, the relationships between observed $O_3$ and $NO_x$ under
different wind conditions at SSI and DT are investigated in this section.
Figure 7 presents the daytime and nighttime $O_3/NO_x$-wind relationships in MAM
(March–May), JJA (June–August), SON (September–November), and DJF (December–
February), respectively during 2012–2017. The SW and W winds were associated with
higher $NO_x$ concentrations in both daytime and nighttime. The result is consistent with the
observed CO changes in Sect. 3.1. Since there is no local anthropogenic emission at SSI,
the higher levels of $NO_x$ and CO were mainly resulted from the transport of more polluted
urban plumes by the SW and W winds. Generally, observed daytime $O_3$ and $NO_x$



concentrations presented opposite variations with the wind direction changes (Fig. 7a). In
SON and DJF, the correlation coefficients (Rs) between daytime $O_3$ and $NO_x$ were -0.72
and -0.75, respectively, indicating that the $O_3$ formation was inhibited by increased $NO_x$
concentrations. The results are in accordance with Tie et al. (2013), who suggested that
the VOC-limited regime of $O_3$ formation was not only confined in urban Shanghai, but also
extended to a broader regional area surrounding Shanghai. However, in MAM and JJA,
the daytime $O_3$-$NO_x$ variations presented totally different patterns under SW and W wind
conditions. As wind directions turned from E-SE to SW-W, observed mean $NO_x$
concentrations increased from about 10 ppbv to 20 ppbv, while observed mean $O_3$
concentrations increased from 50–60 ppbv to 70–80 ppbv. The enhancements in daytime
$O_3$ levels suggested that there should be persistent production of $O_3$ in the polluted air
masses carried by the SW and W winds in MAM and JJA.

Based on observations and WRF-Chem simulations, Tie et al. (2013) suggested

considerable $O_3$ production in aged city plumes in the downwind area of Shanghai. Since
air masses affecting SSI site were directly originated from Shanghai under the SW and W
wind conditions (Fig. 1), the observed $O_3$ enhancements should be mainly attributed to the
$O_3$ production in the city plumes carried by SW and W winds. Studies during the
MIRAGE-Shanghai campaign suggested several factors that contributed to the $O_3$
enhancements in aged city plumes downwind Shanghai. First, as there is a large area of
forest located in the south of Shanghai, Geng et al. (2011) suggested that continuous
oxidation of isoprene emitted by the biogenic sources could result in enhanced production
of hydrogen radicals ($HO_2$) especially in warm seasons. Once the air massed were



transported north and mixed with high $NO_x$ emissions, $O_3$ would be quickly produced.
However, the impacts of biogenic emissions on $O_3$ production were mainly limited in the
south part of Shanghai, which can hardly influence the atmosphere in the SSI region.
Then, Tie et al. (2013) further illustrated that the OH reactivity of alkane, alkene, aromatics,
and oxygenated VOCs (OVOCs) contributed to the $O_3$ formation in city plumes. Among
them, the influence of alkane, alkene and aromatics mostly occurred within or near the city,
while the OVOCs could be produced or emitted during the transport of the city plumes,
resulting in substantial $O_3$ enhancements in aged city plumes at 100–200 km downwind
Shanghai.
The SSI is located approximately 100 km northeast from the downtown area of
Shanghai. In MAM and JJA, the SW and W winds carried air masses with enhanced
OVOCs oxidation and $O_3$ production, resulting in elevated daytime $O_3$ levels on the island.
While in SON and DJF, the observed $O_3$ decreases at SSI during SW and W winds
suggested lower efficiency of $O_3$ productivity in the city plumes. That might be because
that fewer OVOCs were released or produced downwind the city due to the lower
temperature and weaker solar radiation (Cai et al., 2009). In addition, in SON and DJF, the
SW and W winds were usually related to low pressure system with large cloud cover and
rich water vapor in Shanghai, which could also lead to depressed photochemical reactions
and decreased $O_3$ levels. At night, observed $O_3$ and $NO_x$ displayed totally opposite
changes with wind directions (Fig. 7b), indicating $O_3$ depression by nighttime $NO_x$ titration
in all the seasons. High $O_3$ levels were observed under northeasterly, easterly and
southeasterly oceanic wind conditions, ranging from 50–60, 30-55, 55–60, and 40–50





ppbv respectively at night in MAM, JJA, SON, and DJF. More detailed measurements are
still needed to further understand the impacts of city plumes on the $O_3$ levels in oceanic
air.
3.5 Impacts of offshore ozone on urban ozone air quality in Shanghai
As is presented in Sect. 3.2 and 3.3, observed $O_3$ concentrations at SSI were much higher
than those at urban site (XJH), suggesting higher levels of $O_3$ in oceanic air than those on
the continent. Therefore, sea breezes tend to bring more $O_3$ to the continent, aggravating
$O_3$ pollution in coastal cities. Shanghai is one of the largest cities located on the east coast
of China, experiencing severe $O_3$ pollution in recent years (Xu et al., 2019; Gu et al.,
2020). The easterly winds from the ocean greatly affect the Shanghai region, accounting
for 64–78 % of the total flows in non-winter months during the period 2012–2017. To
understand the impacts of higher $O_3$ in oceanic air on the urban air quality, numerical
experiments are conducted using the WRF-Chem model to examine the response of $O_3$
levels in Shanghai to various oceanic air inflow conditions in this section.
Simulations are performed during September 1–30 2014 when the prevailing winds
were mostly northeasterly and easterly in the Shanghai region. The occurrence
frequencies of the northeasterly and easterly winds were 23% and 27% respectively,
during the simulation period, suggesting dominant influence of the oceanic air inflows on
the city of Shanghai. Consistent with above analysis, observed air $O_3$ concentrations were
much higher in oceanic regions than those in city areas, with monthly mean values of 30.9
and 57.7 ppbv, respectively at XJH and SSI in September 2014. The chemical boundary
conditions (BCs) of the regional model can represent the inflows conditions to explore





their impacts on surface concentrations of air pollutants over a certain continent region.
Using this method, Pfiter et al. (2011) proposed that chemical inflows taken from different
observational and model datasets could result in differences of ±15 ppbv in $O_3$ levels in
the US west coast region. Therefore, three sets of numerical experiments are conducted
as follows to access the impacts of oceanic $O_3$ air inflows on the urban $O_3$ air quality in
Shanghai. All the simulations are driven by the same emissions, initial conditions, physical
and chemical schemes.

(1) BC_40: $O_3$ concentrations at the eastern lateral boundary of the domain on the

ocean are assigned to 40 ppbv, which is provided by the MOZART-4 model, closed to the
observed urban $O_3$ levels (29.0–38.4 ppbv) in Shanghai in September. The chemical BCs
are updated every 6 hours.

(2) BC_50: Same as BC_40, but with $O_3$ concentrations setting to 50 ppbv at the

eastern lateral boundary of the domain.

(3) BC_60: Same as BC_40, but with $O_3$ concentrations at the eastern lateral

boundary of the domain setting to 60 ppbv according to the observed $O_3$ levels at SSI
(50.9–71.0 ppbv) in September.

Figure 8 displays the simulated monthly mean distributions of surface $O_3$

concentrations in BC_40, BC_50 and BC_60 scenarios, respectively. The calculated
distributions of $O_3$ agree with observations, which exhibit lower values in urban regions
compared to those in rural and ocean areas, indicating strong $O_3$ depressions in the city of
Shanghai due to the VOC-limited $O_3$ formation regime. The R values between the
simulated and observed $O_3$ concentrations are all larger than 0.50 at XJH, suggesting





good prediction of $O_3$ variations by the model. Table 4 displays the statistical results of the
comparisons between the simulated and observed surface $O_3$ concentrations at SSI and
XJH, respectively. Generally, the WRF-Chem model underestimates $O_3$ concentrations at
both XJH and SSI site. Taken the BC_40 scenario for example, the $O_3$ concentrations are
underestimated by 27% and 36% respectively at XJH and SSI, suggesting larger
underestimation of $O_3$ concentrations in ocean regions. Model results suggest that
increases in the eastern boundary $O_3$ lead to increases in $O_3$ concentrations at both urban
and remote sites when the prevailing winds are mostly easterly in Shanghai. With $O_3$
concentrations increasing from 40 to 60 ppbv in the easterly inflows, the simulated
monthly mean $O_3$ concentrations increase by 7.0 and 10.4 ppbv, respectively at XJH and
SSI. The underestimation of $O_3$ levels by the model is also significantly improved in the
BC_60 scenario, with the chemical BCs of $O_3$ more close to the observed chemical inflow
conditions. Compared to those in the BC_40 scenario, the normalized mean bias (NMBs)
of the predicted $O_3$ concentrations reduced from -36.1 % to -18.1 % at SSI and -27.6% to
-4.6% at XJH in the BC_60 scenario, suggesting a crucial role of the eastern oceanic air
inflows in influencing $O_3$ air quality in Shanghai.

The calculated monthly mean differences in surface $O_3$ concentrations between

simulations in different scenarios are further presented in Fig. 9. Since the dominant winds
are easterly during the simulation period, distinct changes in surface $O_3$ concentrations
throughout Shanghai are generated, exhibiting generally gradient increases from the
ocean to the continent as $O_3$ increases in the oceanic air inflows. With every 10 ppbv
increases in $O_3$ levels in oceanic air, the simulated surface mean $O_3$ concentrations





increase by 3–6 ppbv in the land area and 4–7 ppbv in the offshore region. Due to the
strong $O_3$ depressions associated with high anthropogenic emissions, the simulated $O_3$
enhancements are relatively lower in the central urban region compared to those in
surrounding areas. Even so, simulated mean $O_3$ concentrations still exhibit 6–8 ppbv
increases in the BC_60 scenario, accounting for approximately 30% of the simulated $O_3$
concentrations in the BC_40 case. During the period 2012–2017, most of the measured
$O_3$ concentrations ranged between 50—60 ppbv at SSI in non-winter seasons. Carried by
the easterly inflows, these oceanic air masses with higher $O_3$ levels (50—60 ppbv) could
be transported to the coastal regions, resulting in approximately 20—30% increases in
urban $O_3$ concentrations in Shanghai according to the sensitivity results.
4 Conclusions
In this paper, we present the first relatively long and continuous measurements of oceanic
air $O_3$ conducted at an offshore monitoring station on the Sheshan Island during January 1
2012 to September 15 2017.The southwesterly and westerly winds are proved to carry
more pollutants to the SSI site, exerting greater influence of human activities on the
oceanic atmosphere over the offshore region of the East China Sea. Since the two kinds
of winds exhibited minimum occurrence frequencies and wind speeds in September and
October, atmosphere at SSI during the two months are considered to be less affected by
the transport of regional pollution.
Compared to those in urban (XJH) and rural (DT) sites, the observed $O_3$ levels were
higher at SSI, with mean concentrations of 50.1 ppbv during the observation period.
Similar seasonal and diurnal patterns of $O_3$ were observed at SSI and XJH; however, the



amplitudes of $O_3$ variations were much smaller at the offshore site (SSI). Since $O_3$
formation in Shanghai and its surrounding regions were VOC-limited, the observational
results suggested that the production and depression of $O_3$ could be weaker in the ocean
regions due to weak influence of the anthropogenic emissions. Observed mean
$O_{3-max}/O_{3-min}$ ratios also exhibited lower values at SSI (3.03) than those at XJH (5.20), with
minimum values ranging from 1.61–2.35 in September and October. The result further
illustrated that SSI was seldom affected by the anthropogenic emissions, especially in
September and October.

The multi-year changes of the oceanic $O_3$ at SSI are investigated using the

Mann-Kendall trend test and the Theil-Sen trend estimate method during 2012–2017.
Different from the significant $O_3$ increases detected at XJH and other rural sites reported
in previous studies, the observed mean $O_3$ concentrations at SSI exhibited statistically
insignificant increasing changes (1.12 ppbv $yr^{-1}$, $\alpha>0.10$) during the observation period
and insignificant decreasing changes (-0.72 ppbv $yr^{-1}$, $\alpha>0.10$) in September and October
when the transport of city pollutants had minimum impacts on the island. Due to fewer
impacts of anthropogenic emissions, most of the observed changes in $O_3$ at SSI could be
attributed to the changes of meteorological parameters. Observed wind speed exhibited
significant increases (0.21 m $s^{-1}$, $\alpha>0.10$) in September and October during the
observation period, suggesting that enhanced diffusion conditions could be an important
meteorological factor in determining the decreases in $O_3$ concentrations during the
observation period.

The impacts of urban plumes on $O_3$ levels in oceanic air at SSI are evaluated by





studying the relationships between observed $O_3$ and $NO_x$ under different wind conditions.
The SW and W winds usually carried air masses with higher $NO_x$ concentrations in both
daytime and nighttime to the island. Generally, observed daytime and nighttime $O_3$
concentration decreased as $NO_x$ concentration increases in SW and W winds, exhibiting
typical VOC-limited characteristics of $O_3$ formation. The pattern was more typical in SON
and DJF, with R values of -0.72 and -0.75, respectively between $O_3$ and $NO_x$
concentrations. In MAM and JJA, the daytime $O_3$-$NO_x$ variations presented kind of
positive relationships under SW and W wind conditions, suggesting continuous $O_3$
production in aged city plumes from Shanghai. As reported in previous studies during the
MIRAGE-Shanghai campaign, enhanced OVOCs oxidation should be the most important
driver of the observed $O_3$ enhancements in the city plumes transported by the SW and W
winds.

The influence of the oceanic $O_3$ air inflows on urban $O_3$ air quality in Shanghai are

quantified during an easterly wind dominant episode (September 1–30, 2014). Numerical
experiments are conducted with chemical BCs of $O_3$ assigned according to different inflow
conditions using the WRF-Chem model. Model results suggest that increases of $O_3$ in the
easterly oceanic air inflows will lead to gradient increases from the ocean to the continent.
With every 10 ppbv $O_3$ increases, the calculated surface mean $O_3$ concentrations can
increase by 3–6 ppbv in the land and 4–7 ppbv in the offshore region. Compared to those
in surrounding regions, $O_3$ in central city of Shanghai exhibited lower enhancements in
response to the $O_3$ increases in oceanic air inflows due to strong $O_3$ depression processes.
Even so, the impacts of the oceanic air inflows can still lead to 20—30% increases in





urban $O_3$ concentrations which should be crucially considered in dealing with $O_3$ pollution
in large coastal cities like Shanghai.

*Data availability.* The data used in this paper can be provided upon request from Dr.
Jianming Xu (metxujm@163.com).

*Author contribution.* YG and JX came up with the original idea, designed the analysis
methods, developed the model code, and performed the simulations. WG provided the
observational data. YG and YQ conducted the analysis of the observations and model
results. YG prepared the manuscript with contributions from all co-authors.

*Competing interest.* The authors declare that they have no conflict of interest.

*Acknowledgements.* This work was supported by Shanghai Sailing program
(18YF1421200) and Science and Technology Commission of Shanghai Municipality
(Grand No. 19DZ1205003).



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





Table 1 Mean CO mixing ratios (ppmv) under north (N), northeast (NE), east (E),
southeast (SE), south (S), southwest (SW), west (W), northwest (NW) and calm (C) wind
conditions at Dongtan (DT) site, a remote rural site near the Sheshan Island (SSI) during
2012 to 2017.

|    | N    | NE   | E    | SE   | S    | SW       | W        | NW   | C    |
|----|------|------|------|------|------|----------|----------|------|------|
| CO | 0.31 | 0.27 | 0.25 | 0.23 | 0.27 | **0.44** | **0.56** | 0.38 | 0.34 |





Table 2 Monthly mean wind speeds (m s⁻¹) and occurrence frequencies (%) of the
southwest (SW) and west (W) winds at Dongtan (DT) site, a remote rural site near the
Sheshan Island (SSI) during 2012 to 2017.

|  | Jan. | Feb. | Mar. | Apr. | May | Jun. | Jul. | Aug. | Sep. | Oct. | Nov. | Dec. |
|---|---|---|---|---|---|---|---|---|---|---|---|---|
| SW+W | 11.5 | 9.2 | 11.9 | 13.2 | 12.7 | 9.8 | 17.7 | 10.8 | **6.1** | **5.2** | 11.9 | 15.1 |
| Wind speed | 2.70 | 2.93 | 2.98 | 3.04 | 2.86 | 2.51 | 2.65 | 2.77 | **2.49** | **2.50** | 2.55 | 2.54 |





Table 3 Statistical results of the Mann-Kendall test and Theil-Sen trend estimate for daily
mean values of NO$_x$, CO mixing ratios, temperature (T), and wind speed (WS) in
September and October at Dongtan (DT) site, a remote rural site near Sheshan Island
(SSI) site during the 2012–2017 period. The units of the calculated slopes are ppbv yr$^{-1}$ for
NO$_x$ and CO, °C yr$^{-1}$ for T, and m s$^{-1}$ yr$^{-1}$ for WS.

|  | NO$_x$ | CO | T | WS |
|---|---|---|---|---|
| Slope Estimate | 0.48* | 2.67$^\Delta$ | 0.15$^\Delta$ | 0.21* |

*The result is significant at the 95% confidence level.
$^\Delta$The result cannot pass the Mann-Kendall trend test at the 90% confidence level.





Table 4 Statistical results of the comparisons between the simulated and observed
surface $O_3$ concentrations at Xujiahua (XJH) and Sheshan Island (SSI) sites during
September 2014. The calculated $O_3$ levels are obtained from BC_40, BC_50 and BC_60
simulations, respectively. Values of the average surface $O_3$ concentrations (Mean) and
normalized mean bias (NMB) are displayed. The NMB is defined as NMB=$\frac{\sum_{i=1}^{n}(P_i - O_i)}{\sum_{i=1}^{n} O_i}$,
where $P_i$ and $O_i$ are predicted and observed ozone mixing ratios for sample $i$, $n$ is the
number of total samples (numbers in parentheses).

| | Cases | XJH (641) | SSI (720) |
|---|---|---|---|
| | Observation | 30.4 | 57.7 |
| Mean | BC_40 | 22.0 | 36.9 |
| (ppbv) | BC_50 | 25.1 | 41.8 |
| | BC_60 | 29.0 | 47.3 |
| | BC_40 | -27.6 | -36.1 |
| NMB (%) | BC_50 | -17.5 | -27.5 |
| | BC_60 | -4.6 | -18.1 |



**Figure Captions**

**Figure 1** Land cover of Shanghai and corresponding locations and landscapes of Xujiahui

(XJH, urban), Dongtan (DT, rural) and Sheshan Island (SSI, remote and oceanic) stations.

**Figure 2** Monthly wind rose diagrams averaged over the period of 2012 to 2017 at

Dongtan (DT) site, a remote rural site near the Sheshan Island (SSI).

**Figure 3** Monthly mean CO mixing ratios under north (N), northeast (NE), east (E),

southeast (SE), south (S), southwest (SW), west (W), northwest (NW) and calm (C) wind

conditions at Dongtan (DT) site, a remote rural site near the Sheshan Island (SSI) during

2012 to 2017.

**Figure 4** Monthly and year-round mean diurnal variations of $O_3$ (ppbv) at Sheshan Island

(SSI, remote and oceanic) and Xujiahui (XJH, urban) sites during 2012 to 2017.

**Figure 5** Calculated monthly mean ratios of daily maximum $O_3$ concentrations ($O_{3\text{-max}}$) to

minimum $O_3$ concentrations ($O_{3\text{-min}}$) at Sheshan Island (SSI, remote and oceanic) and

Xujiahui (XJH, urban) sites, respectively during 2012 to 2017.

**Figure 6** Variations of (a) monthly mean $O_3$ concentrations at Sheshan Island (SSI,

remote and oceanic) and Xujiahui (XJH, urban) sites during the period 2012–2017, (c)

corresponding variations of daily mean $O_3$ concentrations at SSI and XJH in September

and October, and (c) variations of mean $O_3$ concentrations during daytime (10:00-16:00

LST) and nighttime (23:00-04:00 LST) at SSI.

**Figure 7** Daytime and nighttime mean $O_3$ mixing ratios (ppbv) at Sheshan Island (SSI)

and $NO_x$ mixing ratios (ppbv) at Dongtan (DT) site, a remote rural site near SSI under

north (N), northeast (NE), east (E), southeast (SE), south (S), southwest (SW), west (W),





and northwest (NW) wind conditions in MAM (March–May), JJA (June–August), SON
(September–November), and DJF (December–February), respectively during 2012 to

731    2017.

**Figure 8** Calculated distributions of monthly mean $O_3$ concentrations (shades, ppbv) from
BC_40, BC_50 and BC_60 simulations, respectively in September 2014. Model results
are compared with observed mean $O_3$ concentrations (circles, ppbv) obtained from
Xujiahui (XJH, urban) and Sheshan Island (SSI, remote and oceanic) sites. Also shown is
the calculated wind field (m s$^{-1}$) averaged over the same period.
**Figure 9** Mean differences in surface $O_3$ concentrations (ppbv) simulated with different
chemical boundaries: (a) BC_50 minus BC_40, (b) BC_60 minus BC_40, and (c) BC_60
minus BC_50 in September 2014. Also shown is the calculated wind field (m s$^{-1}$) averaged
over the simulation period.





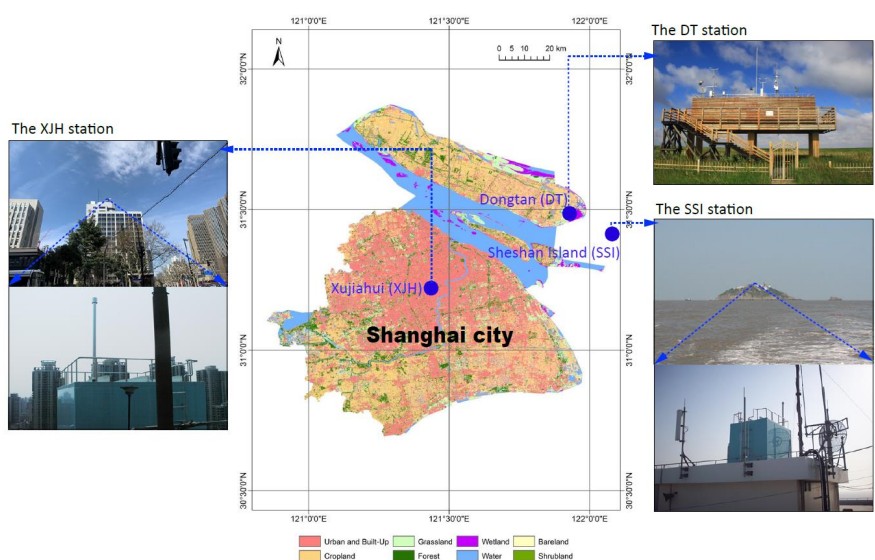


Figure 1 Land cover of Shanghai and corresponding locations and landscapes of Xujiahui
(XJH, urban), Dongtan (DT, rural) and Sheshan Island (SSI, remote and oceanic) stations.



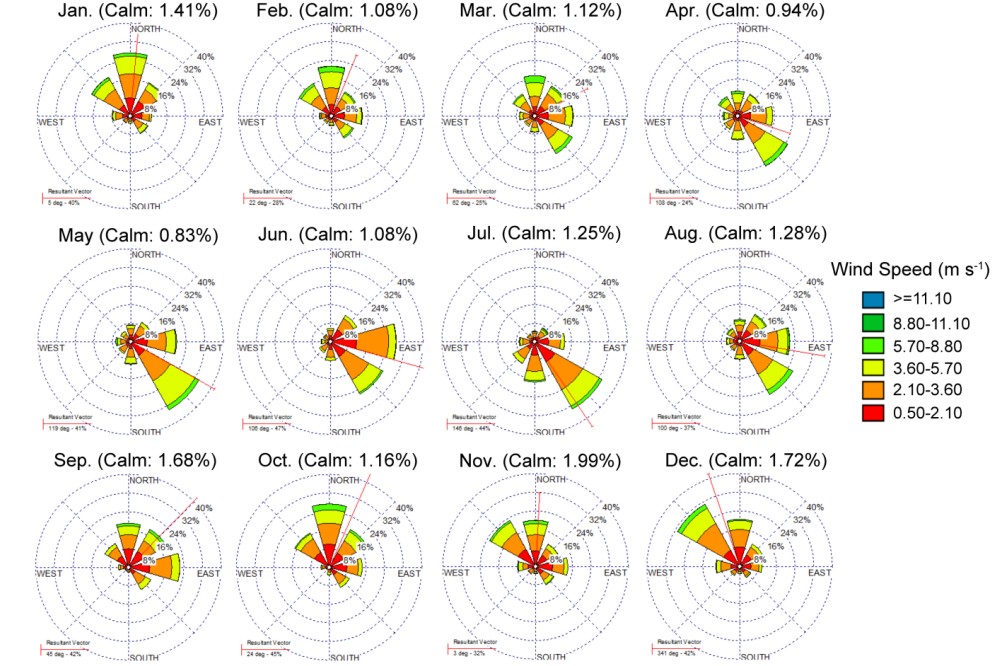


Figure 2 Monthly wind rose diagrams averaged over the period of 2012 to 2017 at
Dongtan (DT) site, a remote rural site near the Sheshan Island (SSI).

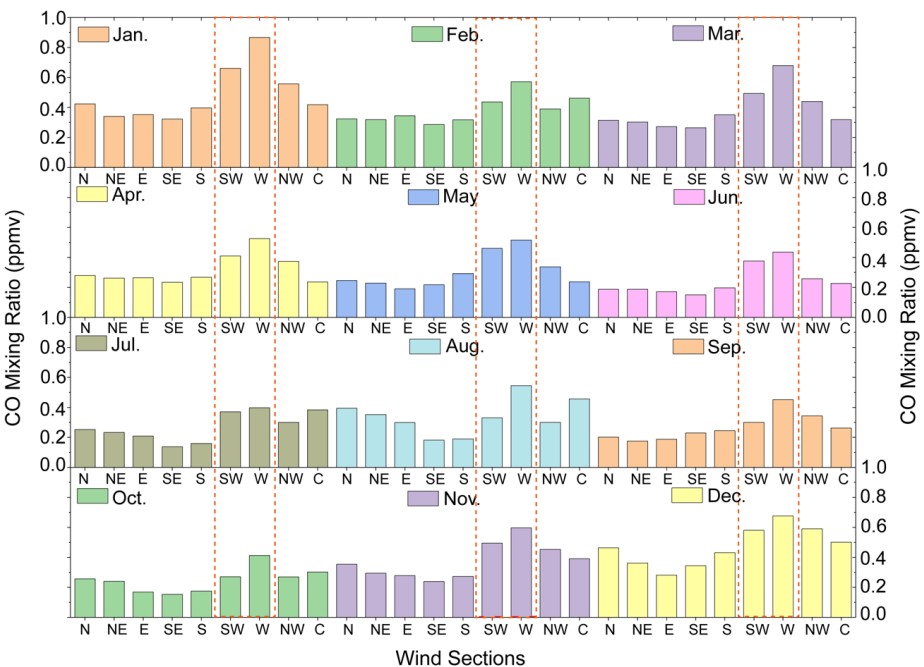

Figure 3 Monthly mean CO mixing ratios under north (N), northeast (NE), east (E),

southeast (SE), south (S), southwest (SW), west (W), northwest (NW) and calm (C) wind

conditions at Dongtan (DT) site, a remote rural site near the Sheshan Island (SSI) during

2012 to 2017.





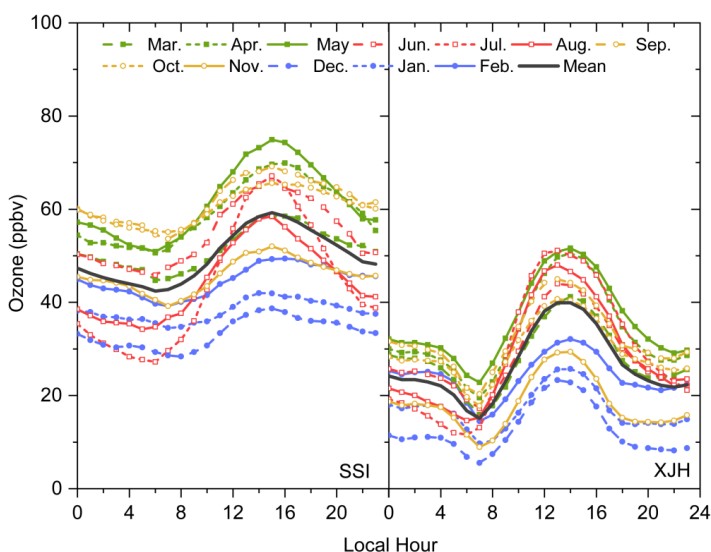

752

Figure 4 Monthly and year-round mean diurnal variations of $O_3$ (ppbv) at Sheshan Island

(SSI, remote and oceanic) and Xujiahui (XJH, urban) sites during 2012 to 2017.

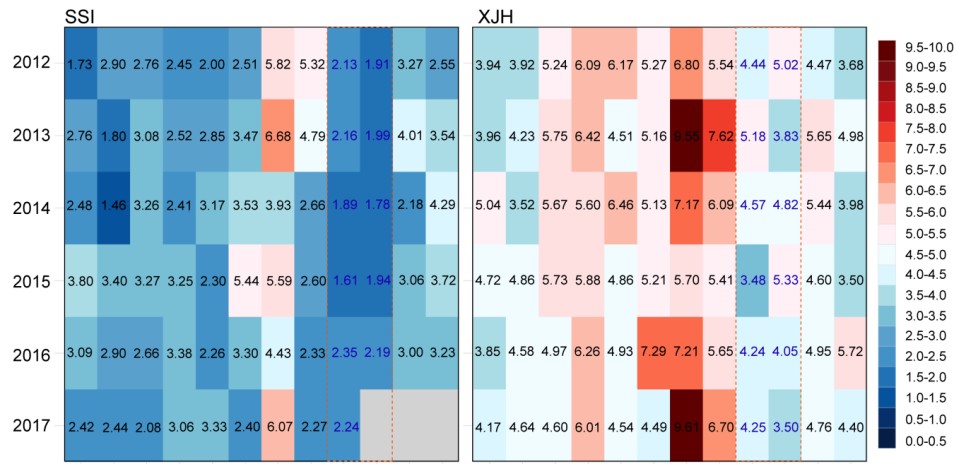

Figure 5 Calculated monthly mean ratios of daily maximum $O_3$ concentrations ($O_{3-max}$) to

minimum $O_3$ concentrations ($O_{3-min}$) at Sheshan Island (SSI, remote and oceanic) and

Xujiahui (XJH, urban) sites, respectively during 2012 to 2017.



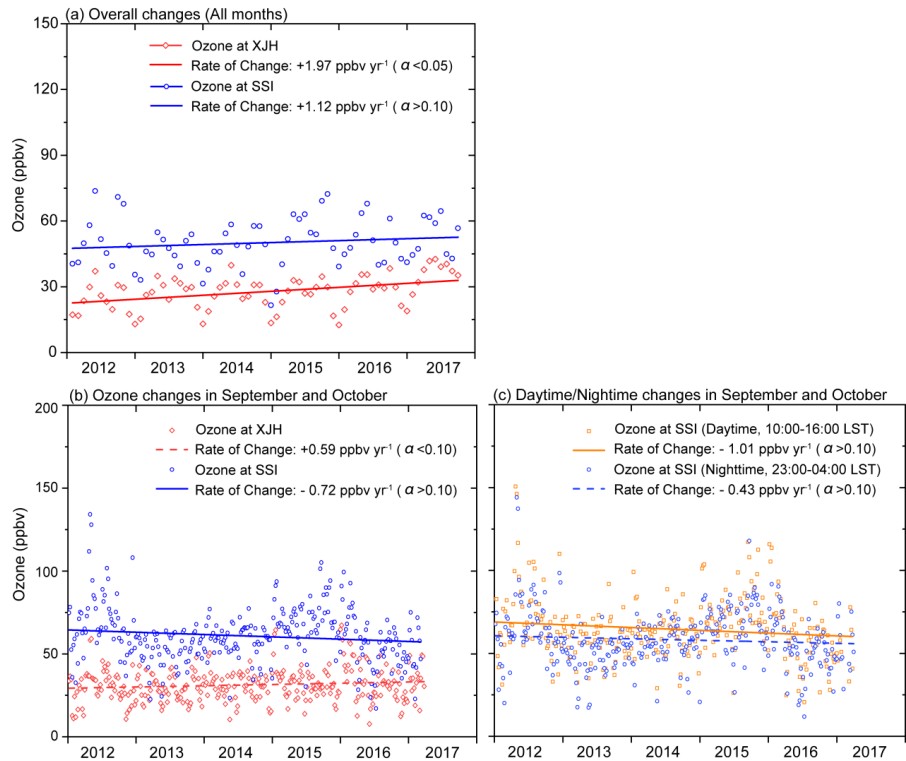

Figure 6 Variations of (a) monthly mean $O_3$ concentrations at Sheshan Island (SSI, remote

and oceanic) and Xujiahui (XJH, urban) sites during the period 2012–2017, (b)

corresponding variations of daily mean $O_3$ concentrations at SSI and XJH in September

and October, and (c) variations of mean $O_3$ concentrations during daytime (10:00-16:00

LST) and nighttime (23:00-04:00 LST) at SSI.





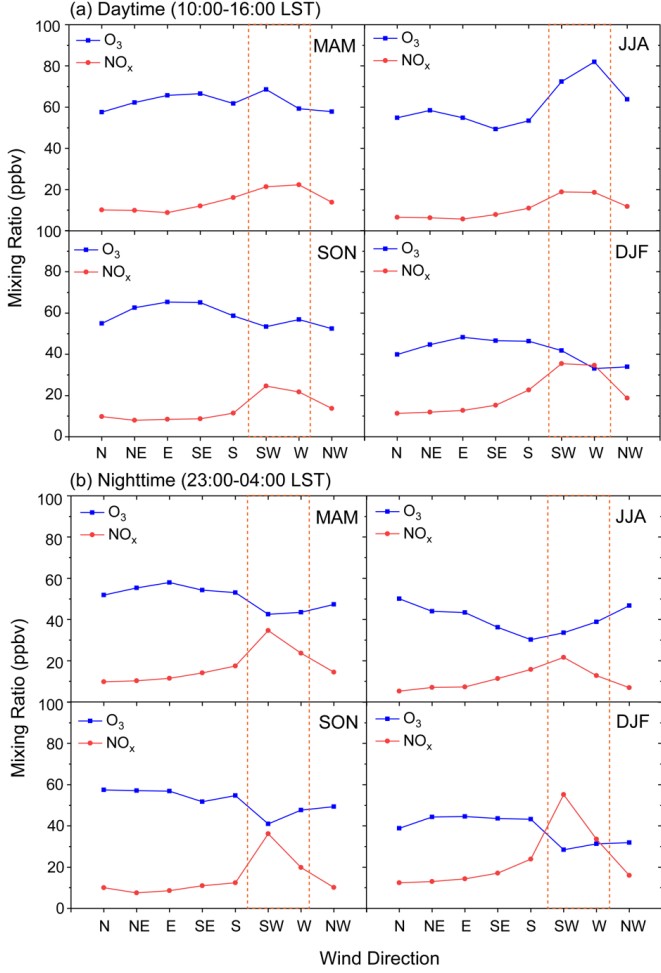

765

Figure 7 Daytime and nighttime mean O$_3$ mixing ratios (ppbv) at Sheshan Island (SSI) and

NO$_x$ mixing ratios (ppbv) at Dongtan (DT) site, a remote rural site near SSI under north (N),

northeast (NE), east (E), southeast (SE), south (S), southwest (SW), west (W), and

northwest (NW) wind conditions in MAM (March–May), JJA (June–August), SON

(September–November), and DJF (December–February), respectively during 2012 to

2017.

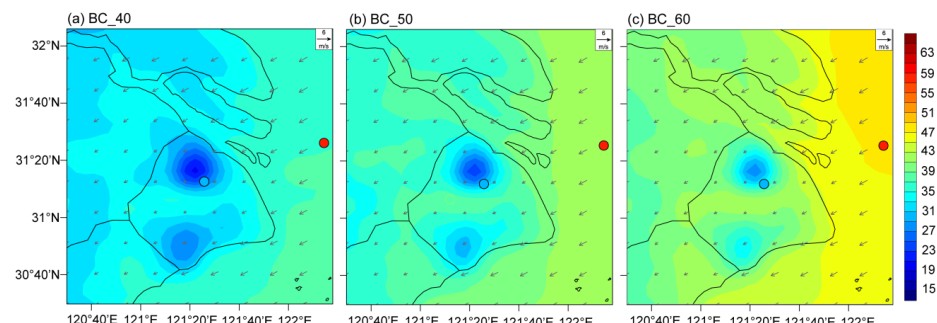


Figure 8 Calculated distributions of monthly mean O$_3$ concentrations (shades, ppbv) from
BC_40, BC_50 and BC_60 simulations, respectively in September 2014. Model results
are compared with observed mean O$_3$ concentrations (circles, ppbv) obtained from
Xujiahui (XJH, urban) and Sheshan Island (SSI, remote and oceanic) sites. Also shown is
the calculated wind field (m s$^{-1}$) averaged over the same period.



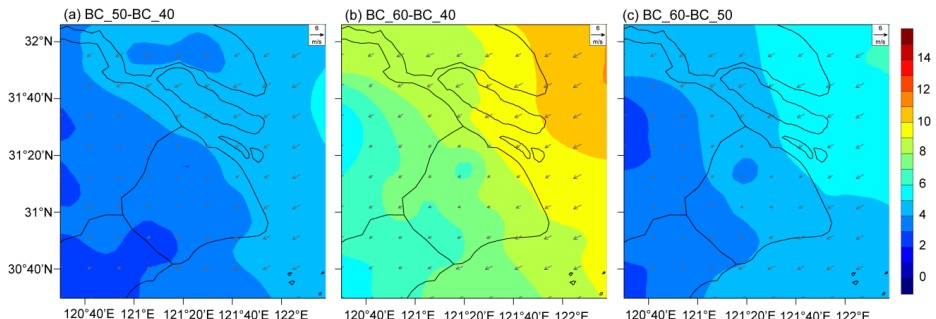


Figure 9 Mean differences in surface $O_3$ concentrations (ppbv) simulated with different

chemical boundaries: (a) BC_50 minus BC_40, (b) BC_60 minus BC_40, and (c) BC_60

minus BC_50 in September 2014. Also shown is the calculated wind field (m s$^{-1}$) averaged

over the simulation period.