# Peer review of "A measurement and model study on ozone characteristics in marine air at a remote"

_Atmospheric Chemistry and Physics, 2020_

## Referee Comment (RC1) · Anonymous Referee #1 · 2 Sep 2020

The manuscript studied the characteristics and changes of baseline ozone in oceanic air in East China based on 6-year continuous measurements conducted at an island site. Corresponding ozone changes under various transport conditions were detailed presented and the impacts of offshore ozone on ozone air quality in Shanghai were quantified using the WRF-Chem model. Since increasing ozone pollution has become an urgent environmental problem in coastal urban agglomerations in East China, the results of this study provide valuable insight into what needs to be considered in dealing with ozone pollution in coastal megacities like Shanghai.

[Figure]

General comments: Results and discussion-Sect.3 presented the overall changes of ozone in oceanic air at SSI. However, the key point was not prominent enough in current version. I suggest focusing more on the novelties of the study (the changes of baseline O3 in oceanic air). Questions of why O3 changes in September and October were analyzed, and what could be the driver of the detected changes need to be deeply reformulated.

Specific comments: 1. Line 59: The location where the increased ozone concentrations were observed should be specified. 2. Line 69: Change "the three" to "those" 3. Line 73: Remove "surface" 4. Line 74: Change "atmospheric oxidation capacity response to" to "atmospheric oxidation capacity of continental air responding to" 5. Line 76: Change "in" to "at" 6. Line 91: Change "covering . . .area" to "covering an area of . . ." 7. Line 93: Change "magnitude" to "levels" 8. Line 120-121: Remove "In addition" 9. Line 125: Please explain the impact 10. Line 158: Add "its" before "surrounding" 11. Line 175: Change "require" to "requires" 12. Line 177: Add "to" before "be" 13. Line 216: Change "cleaner" to "less polluted" 14. Line 228: Change "distinctions" to "variations" 15. Line 234: The observed O3 at DT site need to be provided in the supplementary materials. 16. Line 252: Actually, the study of Scheel et al. (1997) were conducted in Europe. Is 1.4 also a typical value of O3-max/O3-min in Chinese background sites? Please make sure of that. 17. Line 292: How about the trend of O3 observed at DT and Lin'an? 18. Line 298: Change "nearly uncontaminated" to "least contaminated" 19. Line 330: Change "variations of" to "changes in" 20. Line 396-397: Please specify the source of this conclusion. Reference or methods need to be added. 21. Line 456: Do the 6-8 ppbv increases in O3 occur in downtown Shanghai? Please specify it. 22. Line 473: Change "mean concentrations" to "a mean value"

---

## Referee Comment (RC2) · Anonymous Referee #2 · 11 Sep 2020

This study conducted observational and modeling analysis of baseline ozone in oceanic air at Sheshan Island (SSI), which is located to the east of Shanghai city. The authors reported a six-year measurement of ozone concentration at SSI and its ozone level is much high than the value of downtown site in Shanghai. They further highlight the importance of understanding the interaction between urban plume and oceanic inflows in ozone pollution. In particular, their modeling results show that ozone in the oceanic inflows can enhance urban ozone by 20-30%.

Overall, this manuscript is well structured and sites in the scope of this journal. Recent studies have increasingly focused on urban ozone pollution in China, but study on background ozone is still very limited. As such, this study could enrich our understanding of ozone pollution in China, particularly for coastal cities. Although this manuscript is publishable, the current version should be improved in terms of presentation and clarification. I would like the authors to address my following comments.

-This study shows observed ozone levels in a remote site and urban site. In fact, urban ozone in Shanghai are available from Chinese measurement network. It will be great if the authors could have more ozone measurements in this study. For example, Figure 8 is a good place to show more urban ozone data.

- This study gives daily mean of ozone in both observational and modeling calculation. I am wondering if the authors can show more results for MDA8 ozone. Since MDA8 ozone is the standard air quality metric for ozone.

-Line 27: "production" might be more appropriate than "oxidation".

-Line 89: please spell out months.

-Lines 252-253: will this ratio be helpful in this study?

-Line 293: "few" looks not reasonable, since you still saw an increase trend of 1.12ppb yr$^{-1}$.

-Lines 346-348: is there any changes in ozone production sensitivity in response to NOx control?

-Line 397: remove space before "%".

-Line 492: α should be <0.05 according to Table 3.

---

## Author Comment (AC1) · 17 Sep 2020

**Manuscript acp-2020-681**

**Response to Referee #1**

The manuscript studied the characteristics and changes of baseline ozone in oceanic air in East China based on 6-year continuous measurements conducted at an island site. Corresponding ozone changes under various transport conditions were detailed presented and the impacts of offshore ozone on ozone air quality in Shanghai were quantified using the WRF-Chem model. Since increasing ozone pollution has become an urgent environmental problem in coastal urban agglomerations in East China, the results of this study provide valuable insight into what needs to be considered in dealing with ozone pollution in coastal megacities like Shanghai.

We thank the reviewer for all the insightful comments. Please see our point-by-point response (in blue) to the general and specific comments below. The changes that have been made to the manuscript are also listed.

General comments: Results and discussion-Sect.3 presented the overall changes of ozone in oceanic air at SSI. However, the key point was not prominent enough in current version. I suggest focusing more on the novelties of the study (the changes of baseline $O_3$ in oceanic air). Questions of why $O_3$ changes in September and October were analyzed, and what could be the driver of the detected changes need to be deeply reformulated.

Response: Thanks very much for pointing out that. We have rewritten Sect. 3.3 in the revised manuscript as suggested by the reviewer. Please see below:

"As discussed in Sect. 3.1, the prevailing winds carried different levels of pollutants to the SSI, resulting in different impacts on the $O_3$ levels in different months. In September and October, the frequencies of SW and W winds that carried high levels of pollutants were lowest (Table 1–2), exerting least influence on the atmospheric composition at SSI. Therefore, the variations of surface $O_3$ concentrations in September and October at SSI were examined to further assess the changes of least contaminated $O_3$ in the oceanic air. Figure 6b presents the overall changes of daily mean surface $O_3$ concentrations in September and October at SSI and XJH, respectively during the six-year period. The

corresponding mean $O_3$ mixing ratios during the two months were 60.9 and 31.3 ppbv, respectively at SSI and XJH. Compared to the significant elevated $O_3$ concentrations at XJH (0.59 ppbv $yr^{-1}$, $\alpha<0.10$) in September and October, observed $O_3$ at SSI during same months exhibited insignificant decreasing changes from 2017–2017. The changes (-0.72 ppbv $yr^{-1}$, $\alpha>0.10$) were somewhat different from the overall $O_3$ changes (+1.12 ppbv $yr^{-1}$, $\alpha>0.10$) at SSI, suggesting different causes of the observed $O_3$ changes in the oceanic air during September and October.

To investigate possible drivers of the observed changes in the least contaminated $O_3$ in September and October at SSI, Table 3 displays the statistical results of the MK test and Theil-Sen trend estimate for $NO_x$ and CO mixing ratios, temperature, and wind speed during the 2012–2017 period. Statistically significant upward trends were detected in wind speed, with estimated increasing rates of 0.21 m $s^{-1}$ $yr^{-1}$ during the observation period ($\alpha<0.05$). The significantly enhanced surface wind speeds were conducive to the diffusion of $O_3$, which might be an important meteorological driver of the observed decreasing changes in $O_3$ levels at SSI from 2012 to 2017. Observed $NO_x$ and CO levels exhibited increases of 0.48 ppbv $yr^{-1}$ ($\alpha<0.05$) and 2.67 ppbv $yr^{-1}$ ($\alpha>0.10$), respectively in September and October during the six-year period, indicating enhanced transport of pollutants to the oceanic area. Tie et al. (2013) suggested that the VOC-limited regime of $O_3$ formation was not only confined in urban Shanghai, but also extended to a broader regional area surrounding Shanghai. Thus, the elevated $NO_x$ concentrations might not only retard daytime $O_3$ production but also enhance nighttime $O_3$ depression at SSI. Figure 6c further presents corresponding variations of daytime (10:00-16:00 LST) and nighttime (23:00-04:00 LST) mean $O_3$ concentrations at SSI. Both daytime and nighttime $O_3$ concentrations exhibited downward changes, reflecting the $O_3$ response to the enhanced $O_3$ diffusion and depression in September and October. Therefore, the enhanced diffusion and depression of $O_3$ induced by the elevated wind speed and $NO_x$ concentrations might be important causes of the observed $O_3$ changes in September and October at SSI. It should be noted that the influence of radiation cannot be analyzed since observations of solar radiation were not available during the study period. Therefore, more

measurements are still needed to further understand the $O_3$ changes and corresponding drivers in the oceanic air."

Specific comments:

1. Line 59: The location where the increased ozone concentrations were observed should be specified.

Response: We have revised the sentence as "Based on 14-year observations at a coastal site in Hong Kong, Wang et al. (2009) pointed out that enhanced pollution flow from the upwind coastal regions contributed to most of the observed $O_3$ increases in the background atmosphere of South China during 1994–2007. And the increase in background $O_3$, in turn, made a strong contribution of 81% to the increasing rate of $O_3$ in urban Hong Kong.".

2. Line 69: Change "the three" to "those"

Response: Changed.

3. Line 73: Remove "surface"

Response: Removed.

4. Line 74: Change "atmospheric oxidation capacity response to" to "atmospheric oxidation capacity of continental air responding to"

Response: Changed.

5. Line 76: Change "in" to "at"

Response: Changed.

6. Line 91: Change "covering…area" to "covering an area of…"

Response: Changed.

7. Line 93: Change "magnitude" to "levels"

Response: Changed.

8. Line 120-121: Remove "In addition"

Response: Removed.

9. Line 125: Please explain the impact

Response: This part has been revised as "The DT site was set up in a national nature reserve near the coast of Shanghai, where the observed pollutant levels have been reported to well reflect the impacts of megacities in the Yangtze River Delta (YRD) region on the remote atmosphere during the MIRAGE-Shanghai (Megacities Impact on Regional and Global Environment at Shanghai) field campaign".

10. Line 158: Add "its" before "surrounding"

Response: Added.

11. Line 175: Change "require" to "requires"

Response: Changed.

12. Line 177: Add "to" before "be"

Response: Added.

13.Line 216: Change "cleaner" to "less polluted"

Response: Changed.

14. Line 228: Change "distinctions" to "variations"

Response: Changed.

15. Line 234: The observed $O_3$ at DT site need to be provided in the supplementary

materials.

Response: The mean diurnal variations of $O_3$ at DT during the period 2012–2017 have been added to the supplementary materials as Fig. S1. Please see below.

[Figure]

Fig. S1. Mean diurnal variations of $O_3$ at Sheshan Island (SSI, remote and oceanic), Dongtan (DT, rural), and Xujiahui (XJH, urban) station during the period 2012–2017.

16. Line 252: Actually, the study of Scheel et al. (1997) was conducted in Europe. Is 1.4 also a typical value of $O_{3-max}/O_{3-min}$ in Chinese background sites? Please make sure of that.

Response: Thanks for pointing out that. The following description has been added to the second paragraph of Sect. 3.2:" For regional background sites in China, the typical values of $O_{3-max}/O_{3-min}$ were usually in the range of 2–3 (Xu et al., 2008; Meng et al., 2009; Gu et al., 2020). In Lin'an, a continental background site in YRD region, the ratio was reported to increase as a result of $NO_x$ emission changes during past decades, which could reach above 6 during summertime (Xu et al., 2008).".

17. Line 292: How about the trend of $O_3$ observed at DT and Lin'an?

Response: Compared to those at SSI, observed extreme values of $O_3$ concentrations at DT and Lin'an were reported to exhibit more statistically significant (α<0.05) changes

response to the changes of anthropogenic emissions (e.g. $NO_x$) in past decades (Xu et al., 2008; Gao et al., 2017; Gu et al., 2020). We have revised this part in the manuscript.

18. Line 298: Change "nearly uncontaminated" to "least contaminated"

Response: Changed.

19. Line 330: Change "variations of" to "changes in"

Response: Changed.

20. Line 396-397: Please specify the source of this conclusion. Reference or methods need to be added.

Response: We have added Fig. S2 to the supplementary materials and revised this part as :" According to the cluster analysis results (Fig. S2), easterly winds from the ocean greatly affected the Shanghai region, accounting for 64–78% of the total flows in non-winter months during the period 2012–2017."

[Figure]

Fig. S2. Seasonal variations of the 72-h air mass backward trajectories arriving at the Sheshan Island (SSI) site using the Hybrid Single-Particle Lagrangian Integrated Trajectory (HYSPLIT) model (Version 4, Draxler and Hess, 1998) driven by NCEP/NCAR Global Reanalysis Data (2.5°×2.5°). Trajectory clusters for MAM (March–May, left up), JJA (June–August, right up), SON (September–November, left bottom), and DJF (December–February, right bottom) were calculated based on the trajectories of 2012–2017 with steps of 12 h. The corresponding percentage occurrence values for different groups are presented as numbers in black squares.

21. Line 456: Do the 6-8 ppbv increases in $O_3$ occur in downtown Shanghai? Please specify it.

Response: This sentence has been revised as "Even so, simulated mean $O_3$

concentrations still exhibit 6–8 ppbv increases in downtown Shanghai in the BC_60 scenario, accounting for approximately 30% of the simulated $O_3$ concentrations in the BC_40 case.".

22. Line 473: Change "mean concentrations" to "a mean value"

Response: Changed.

Reference

Draxler, R. R. and Hess, G. D.: An overview of the HYSPLIT 4 modelling system for trajectories, despersion, and deposition, Austral. Meterol. Mag., 47, 295–308, 1998.

Gao, W., Tie, X., Xu, J., Huang, R., Mao, X., Zhou, G., and Chang, L.: Long-term trend of $O_3$, in a mega city (Shanghai), China: characteristics, causes, and interactions with precursors, Sci. Total Environ., 603–604, 425–433, 2017.

Gu, Y., Li, K., Xu, J., Liao, H., Zhou, G.: Observed dependence of surface ozone on increasing temperature in Shanghai, China. Atmos. Environ., 221, 117108, 2020.

Meng, Z. Y., Xu, X. B., Yan, P., Ding, G. A., Tang, J., Lin, W. L., Xu, X. D., and Wang, S. F.: Characteristics of trace gaseous pollutants at a regional background station in Northern China, Atmos. Chem. Phys., 9, 927–936, https://doi.org/10.5194/acp-9-927-2009, 2009.

Tie, X., Geng, F., Guenther, A., Cao, J., Greenberg, J., Zhang, R., Apel, E., Li, G., Weinheimer, A., Chen, J., and Cai, C.: Megacity impacts on regional ozone formation: observations and WRF-Chem modeling for the MIRAGE-Shanghai field campaign, Atmos. Chem. Phys., 13, 5655–5669, 2013.

Wang, T., Wei, X. L., Ding, A. J., Poon, C. N., Lam, Y. S., Li, Y. S., Chan, L. Y., and Anson, M.: Increasing surface ozone concentrations in the background atmosphere of Southern China, 1994–2007, Atmos. Chem. Phys., 9, 6217–6227, 2009.

Xu, X., Lin, W., Wang, T., Yan, P., Tang, J., Meng, Z., and Wang, Y.: Long-term trend of surface ozone at a regional background station in eastern China 1991–2006: enhanced variability, Atmos. Chem. Phys., 8, 2595–2607, 2008.

---

## Author Comment (AC2) · 17 Sep 2020

**Manuscript acp-2020-681**

**Response to Referee #2**

This study conducted observational and modeling analysis of baseline ozone in oceanic air at Sheshan Island (SSI), which is located to the east of Shanghai city. The authors reported a six-year measurement of ozone concentration at SSI and its ozone level is much high than the value of downtown site in Shanghai. They further highlight the importance of understanding the interaction between urban plume and oceanic inflows in ozone pollution. In particular, their modeling results show that ozone in the oceanic inflows can enhance urban ozone by 20-30%.

We thank the reviewer for all the insightful comments. Please see our point-by-point response (in blue) to the general and specific comments below. The changes that have been made to the manuscript are also listed.

Overall, this manuscript is well structured and sites in the scope of this journal. Recent studies have increasingly focused on urban ozone pollution in China, but study on background ozone is still very limited. As such, this study could enrich our understanding of ozone pollution in China, particularly for coastal cities. Although this manuscript is publishable, the current version should be improved in terms of presentation and clarification. I would like the authors to address my following comments.

-This study shows observed ozone levels in a remote site and urban site. In fact, urban ozone in Shanghai are available from Chinese measurement network. It will be great if the authors could have more ozone measurements in this study. For example, Figure 8 is a good place to show more urban ozone data.

Response: Thanks for the suggestions. Since $O_3$ measurements in urban Shanghai from Chinese measurement network are only publicly available after March 2013, the data length cannot cover the whole study period in this study. To provide more measurements in Fig. 8, we added continuous $O_3$ measurements obtained from Sheshan (SS), Pudong (PD), and DT (Dongtan) sites in Shanghai. $O_3$ concentrations were measured using the

same method as those at SSI and XJH during the period 2012–2017. Figure 8 and Table 4 have been revised same as Fig. R1 and Table R1. Corresponding discussions in Sect. 3.5 are revised as follows:

"Figure 8 displays the simulated and observed monthly mean distributions of surface $O_3$ concentrations in BC_40, BC_50 and BC_60 scenarios, respectively. In addition to the observations at XJH and SSI, $O_3$ measurements obtained from other three sites, Pudong (PD, suburban), Sheshan (SS, rural), and Dongtan (DT, rural), during the same period were introduced to evaluate the model's performance in simulating $O_3$ in Shanghai. The $O_3$ concentrations at all the sites were measured using the same method as described in Sect. 2.1. The calculated distributions of $O_3$ agree with observations, which exhibit lower values in urban regions compared to those in rural and ocean areas, indicating strong $O_3$ depressions in the city of Shanghai due to the VOC-limited $O_3$ formation regime. The R values between the simulated and observed $O_3$ concentrations are all larger than 0.50 at continental sites (XJH, PD, SS, and DT), suggesting good prediction of $O_3$ variations by the model.

Table 4 displays the statistical results of the comparisons between the simulated and observed surface $O_3$ concentrations at different sites in Shanghai. Generally, the WRF-Chem model underestimates $O_3$ concentrations at all the sites in most cases. Taken the BC_40 scenario for example, the $O_3$ concentrations are underestimated by 9.4–27.6% at continental sites and 36.1% at SSI, suggesting larger underestimation of $O_3$ concentrations in oceanic regions. Model results further suggest that elevated $O_3$ levels in the eastern chemical BCs would lead to increases in $O_3$ concentrations at both urban and remote sites when the prevailing winds are mostly easterly in Shanghai. With $O_3$ concentrations increasing from 40 to 60 ppbv in the easterly oceanic air inflows, the simulated monthly mean $O_3$ concentrations increase by 7.0–9.7 ppbv at continental sites and 10.4 ppbv at SSI. The underestimation of $O_3$ levels by the model is also greatly improved in the BC_60 scenario, when the chemical BCs of $O_3$ are more close to the observations. Compared to those in the BC_40 scenario, the normalized mean bias (NMBs) of the predicted $O_3$ concentrations reduced at most sites in the BC_60 scenario,

for example from -36.1 % to -18.1 % at SSI and -27.6% to -4.6% at XJH, suggesting a crucial role of the eastern oceanic air inflows in influencing $O_3$ air quality in Shanghai."

[Figure]

Figure R1 Calculated distributions of monthly mean $O_3$ concentrations (shades, ppbv) from BC_40, BC_50 and BC_60 simulations, respectively in September 2014. Model results are compared with observed mean $O_3$ concentrations (circles, ppbv) obtained from Sheshan (SS), Xujiahua (XJH), Pudong (PD), DT (Dongtan) and Sheshan Island (SSI) sites. Also shown is the calculated wind field (m s$^{-1}$) averaged over the same period.

Table R1 Statistical results of the comparisons between the simulated and observed surface $O_3$ concentrations at Sheshan (SS), Xujiahua (XJH), Pudong (PD), DT (Dongtan) and Sheshan Island (SSI) sites during September 2014. The calculated $O_3$ levels are obtained from BC_40, BC_50 and BC_60 simulations, respectively. Values of the average surface $O_3$ concentrations (Mean) and normalized mean bias (NMB) are displayed. The NMB is defined as NMB=$\frac{\sum_{i=1}^{n}(P_i - O_i)}{\sum_{i=1}^{n} O_i}$, where $P_i$ and $O_i$ are predicted and observed ozone mixing ratios for sample $i$, $n$ is the number of total samples (numbers in parentheses).

|  | Cases | SS (681) | XJH (641) | PD (690) | DT (690) | SSI (720) |
|---|---|---|---|---|---|---|
|  | Observation | 39.7 | 30.4 | 40.3 | 46.4 | 57.7 |
| Mean | BC_40 | 36.0 | 22.0 | 29.5 | 35.3 | 36.9 |
| (ppbv) | BC_50 | 39.1 | 25.1 | 33.3 | 39.6 | 41.8 |
|  | BC_60 | 43.1 | 29.0 | 37.9 | 45.0 | 47.3 |
| NMB(%) | BC_40 | -9.4 | -27.6 | -26.7 | -23.9 | -36.1 |
|  | BC_50 | -1.5 | -17.5 | -17.2 | -14.5 | -27.5 |

| | BC_60 | 8.6 | -4.6 | -5.9 | -3.0 | -18.1 |
| --- | --- | --- | --- | --- | --- | --- |

- This study gives daily mean of ozone in both observational and modeling calculation. I am wondering if the authors can show more results for MDA8 ozone. Since MDA8 ozone is the standard air quality metric for ozone.

Response: Thanks for the suggestions. We have presented results for MDA8 $O_3$ in the revised manuscript. The new results added are as follows:

Sect. 3.2:" The observed mean daily maximum 8-h average (MAD8) $O_3$ concentrations exhibited same differences between the two sites, which were 40.1 and 62.0 ppbv, respectively at XJH and SSI."

Sect. 3.3:" The monthly mean MDA8 and daily extreme values of $O_3$ exhibited similar differences between the two sites. The calculated increasing rates of MDA8 $O_3$, $O_{3-max}$ and $O_{3-min}$ were 2.73, 2.77, and 1.35 ppbv yr$^{-1}$ ($\alpha < 0.05$), respectively at XJH, and 1.01, 1.35, and 1.27 ppbv yr$^{-1}$ ($\alpha > 0.10$), respectively at SSI."

-Line 27: "production" might be more appropriate than "oxidation".

Response: Revised.

-Line 89: please spell out months.

Response: Revised.

-Lines 252-253: will this ratio be helpful in this study?

Response: The ratio of daily maximum $O_3$ concentration ($O_{3-max}$) to minimum $O_3$ concentration ($O_{3-min}$) was usually regarded as an indicator to identify if a site could be considered as a typical background site as suggested in previous studies (Cvitas and Klasinc 1993; Cvitas et al., 1995; Vingarzan, 2004). For SSI, the mean ratio of $O_{3-max}$/$O_{3-min}$ was calculated to be 3.03 during the study period, which was consistent with the typical values observed at continental background sites in China (Xu et al., 2008; Meng et al., 2009; Gu et al., 2020). In September and October, the ratio exhibited even low values

at SSI, ranging from 1.61–2.35. The results helped to further indicate that the observed $O_3$ at SSI, especially in September and October, were least contaminated by regional pollution. And the SSI site could be regarded as a typical oceanic background site, providing a good proxy to study the baseline oxidation capacity of oceanic atmosphere in eastern China.

-Line 293: "few" looks not reasonable, since you still saw an increase trend of 1.12ppb yr$^{-1}$.

Response: Thanks for pointing out that. This part has been revised as :"… the statistically insignificant changes of $O_3$ detected at SSI indicated that $O_3$ in the oceanic air remained a constant level during the study period and was less influenced by the decreases of $NO_x$ emissions."

-Lines 346-348: is there any changes in ozone production sensitivity in response to NOx control?

Response: Yes. Based on measurements obtained from the same sites, Xu et al. (2019) has carefully examined the response of $O_3$ production sensitivity to $NO_x$ reductions in Shanghai during the past decade. The $O_3$ isopleth diagram (Fig. R2) constructed by the Ozone Isopleth Plotting Package Research (OZIPR) model suggested that the $O_3$ production had moved from strong VOC-limited regime to slight VOC-limited regime in both urban and suburban sites in Shanghai due to the significant $NO_x$ reductions and slight VOCs changes from 2009 to 2015. In 2017, the observed mean $NO_x$ concentrations at XJH and PD decreased by 3.5 and 0.3 ppbv, respectively compared to those (33.2 ppbv at XJH and 22.6 ppbv at PD) in 2015, indicating that the $O_3$ production could still be VOC-limited in Shanghai according to Fig. S1. The results suggested that the $O_3$ production remained VOC-limited in Shanghai during the study period and the changes in $O_3$ production induced by $NO_x$ reductions did not affect the main conclusion of the manuscript.

[Figure]

Figure R2 The $O_3$ isopleth diagram constructed by the OZIPR model in Shanghai (Xu et al., 2019).

We have revised this part as:" The results are in accordance with Tie et al. (2013) and Xu et al. (2019), who suggested that Shanghai and a broader regional area surrounding the city were all in the VOC-limited $O_3$ formation regime during the study period."

-Line 397: remove space before "%".

Response: Removed.

-Line 492: α should be <0.05 according to Table 3.

Response: Revised.

Reference

Cvitas, T., and Klasinc, L.: Measurement of tropospheric ozone in the Eastern Mediterranean, Boll. Geofisico, 16, 521–527, 1993.

Cvitas, T., Kezele, N., Klasinc, L., and Lisac, J.: Tropospheric ozone measurements in Croatia, Pure Appl. Chem., 67, 1450–1453,1995.

Gu, Y., Li, K., Xu, J., Liao, H., Zhou, G.: Observed dependence of surface ozone on increasing temperature in Shanghai, China. Atmos. Environ., 221, 117108, 2020.

Meng, Z. Y., Xu, X. B., Yan, P., Ding, G. A., Tang, J., Lin, W. L., Xu, X. D., and Wang, S. F.: Characteristics of trace gaseous pollutants at a regional background station in Northern China, Atmos. Chem. Phys., 9, 927–936, https://doi.org/10.5194/acp-9-927-2009, 2009.

Vingarzan, R.: A review of surface ozone background levels and trends, Atmos. Environ., 38, 3431–3442, 2004.

Xu, J., Tie, X., Gao, W., Lin, Y., and Fu, Q.: Measurement and model analyses of the ozone variation during 2006 to 2015 and its response to emission change in megacity Shanghai, China, Atmos. Chem. Phys., 19, 9017–9035, 2019.

Xu, X., Lin, W., Wang, T., Yan, P., Tang, J., Meng, Z., and Wang, Y.: Long-term trend of surface ozone at a regional background station in eastern China 1991–2006: enhanced variability, Atmos. Chem. Phys., 8, 2595–2607, 2008.